# Motion-X: A Large-scale 3D Expressive Whole-body Human Motion Dataset

**Jing Lin**[1,2*‡], **Ailing Zeng**[1*†], **Shunlin Lu**[1,3*‡],
**Yuanhao Cai**[2], **Ruimao Zhang**[3], **Haoqian Wang**[2], **Lei Zhang**[1]
[1]International Digital Economy Academy (IDEA)
[2]Tsinghua University, [3]The Chinese University of Hong Kong, Shenzhen
https://motion-x-dataset.github.io

## Abstract

In this paper, we present Motion-X, a large-scale 3D expressive whole-body motion dataset. Existing motion datasets predominantly contain body-only poses, lacking facial expressions, hand gestures, and fine-grained pose descriptions. Moreover, they are primarily collected from limited laboratory scenes with textual descriptions manually labeled, which greatly limits their scalability. To overcome these limitations, we develop a whole-body motion and text annotation pipeline, which can automatically annotate motion from either single- or multi-view videos and provide comprehensive semantic labels for each video and fine-grained whole-body pose descriptions for each frame. This pipeline is of high precision, cost-effective, and scalable for further research. Based on it, we construct Motion-X, which comprises 15.6M precise 3D whole-body pose annotations (i.e., SMPL-X) covering 81.1K motion sequences from massive scenes. Besides, Motion-X provides 15.6M frame-level whole-body pose descriptions and 81.1K sequence-level semantic labels. Comprehensive experiments demonstrate the accuracy of the annotation pipeline and the significant benefit of Motion-X in enhancing expressive, diverse, and natural motion generation, as well as 3D whole-body human mesh recovery.

## 1 Introduction

Human motion generation aims to automatically synthesize natural human movements. It has wide applications in robotics, animation, games, and generative creation. Given a text description or audio command, motion generation can be controllable to obtain the desired human motion sequence. Text-conditioned motion generation has garnered increasing attention in recent years since it behaves in a more natural interactive way [1, 2, 3, 4, 5, 6, 7, 8, 9, 10].

Although existing text-motion datasets [4, 11, 6, 8] have greatly facilitated the development of motion generation [2, 12, 13, 14, 9], their scale, diversity, and expressive capability remain unsatisfactory. Imagine generating "*a man is playing the piano happily*", as depicted in Fig. 1(a), the motion from existing dataset [4] only includes the body movements, without finger movements or facial expressions. The missing hand gestures and facial expressions severely hinder the high level of expressiveness and realism of the motion. Additionally, certain specialized motions, such as high-level skiing, aerial work, and riding are challenging to be captured in indoor scenes. To sum up, existing datasets suffer from four main limitations: 1) body-only motions without facial expressions and hand poses; 2) insufficient diversity and quantity, only covering indoor scenes; 3) lacking diverse and long-term motion sequences; and 4) manual text labels that are unscalable, unprofessional and labor-intensive. These limitations hinder existing generation methods to synthesize expressive whole-body motion with diverse action types. Therefore, *how to collect large-scale whole-body motion and text annotations from multi-scenarios are critical in addressing the data scarcity issue.*

---

[*]Equal Contribution, [‡] Work done during an internship at IDEA
[†]Corresponding author

37th Conference on Neural Information Processing Systems (NeurIPS 2023) Track on Datasets and Benchmarks.

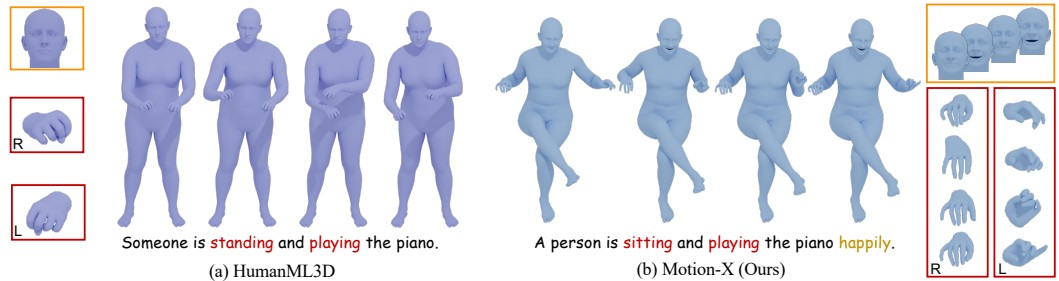

Figure 1: Different from (a) previous motion dataset [4, 8], (b) our dataset captures body, facial expressions, and hand gestures. We highlight the comparisons of facial expressions and hand gestures.

Compared to indoor marker-based mocap systems, markerless vision-based motion capture methods [15, 16, 17, 18, 19, 20] become promising to capture large-scale motions from massive videos. Meanwhile, human motion can be regarded as a sequence of kinematic structures, which can be automatically translated into pose scripts using rule-based techniques [3]. More importantly, although markerless capture (e.g., pseudo labels) is not as precise as marker-based methods, collecting massive and informative motions, especially local motions, could still be beneficial [21, 15, 22, 23, 24]. Besides, text-driven motion generation task requires semantically corresponding motion labels instead of vertex-corresponding mesh labels, and thus have a higher tolerance of motion capture error. Bearing these considerations in mind, we design a scalable and systematic pipeline for motion and text annotation in both multi-view and single-view videos. Firstly, we gather and filter massive video recordings from a variety of scenes with challenging, high-quality, multi-style motions and sequence-level semantic labels. Subsequently, we estimate and optimize the parameters of the SMPL-X model [25] for the whole-body motion annotation. Due to the depth ambiguity and various challenges in different scenes, existing monocular estimation models typically fail to yield satisfactory results. To address this issue, we systematically design a high-performance framework incorporating several innovative techniques, including a hierarchical approach for whole-body keypoint estimation, a score-guided adaptive temporal smoothing and optimization scheme, and a learning-based 3D human model fitting process. By integrating these techniques, we can accurately and efficiently capture the ultimate 3D motions. Finally, we design an automatic algorithm to caption frame-level descriptions of whole-body poses. We obtain the body and hand scripts by calculating spatial relations among body parts and hand fingers based on the SMPL-X parameters and extract the facial expressions with an emotion classifier. We then aggregate the low-level pose information and translate it into textual pose descriptions.

Based on the pipeline, we collect a large-scale whole-body expressive motion dataset named *Motion-X*, which includes 15.6M frames and 81.1K sequences with precise 3D whole-body motion annotations, pose descriptions, and semantic labels. To compile this dataset, we collect massive videos from the Internet, with a particular focus on game and animation motions, professional performance, and diverse outdoor actions. Additionally, we incorporated data from eight existing action datasets [26, 27, 28, 11, 29, 30, 31, 32]. Using *Motion-X*, we build a benchmark for evaluating several state-of-the-art (SOTA) motion generation methods. Comprehensive experiments demonstrate the benefits of *Motion-X* for diverse, expressive, and realistic motion generation (shown in Fig. 1 (b)). Furthermore, we validate the versatility and quality of *Motion-X* on the whole-body mesh recovery task.

Our contributions can be summarized as follows:

- We propose a large-scale expressive motion dataset with precise 3D whole-body motions and corresponding sequence-level and frame-level text descriptions.
- We elaborately design a automatic motion and text annotation pipeline, enabling efficient capture of high-quality human text-motion data at scale.
- Comprehensive experiments demonstrate the accuracy of the motion annotation pipeline and the benefits of *Motion-X* in 3D whole-body motion generation and mesh recovery tasks.

## 2 Preliminary and Related Work

In this section, we focus on introducing existing **datasets** for human motion generation. For more details about the motion generation methods, please refer to the appendix.

| Dataset | Motion Annotation | | | | Text Annotation | | | Scene | | |
|---|---|---|---|---|---|---|---|---|---|---|
| | Clip | Hour | Whole-body? | Source | Motion | Pose | Whole-body? | Indoor | Outdoor | RGB |
| KIT-ML'16 [6] | 3911 | 11.2 | ✗ | Marker-based MoCap | 6278 | 0 | ✗ | ✓ | ✗ | ✗ |
| AMASS'19 [11] | 11265 | 40.0 | ✗ | Marker-based MoCap | 0 | 0 | ✗ | ✓ | ✗ | ✗ |
| BABEL'21 [8] | 13220 | 43.5 | ✗ | Marker-based MoCap | 91408 | 0 | ✗ | ✓ | ✗ | ✗ |
| Posescript'22 [3] | - | - | ✗ | Marker-based MoCap | 0 | 120k | ✗ | ✓ | ✗ | ✗ |
| HumanML3D'22 [4] | 14616 | 28.6 | ✗ | Marker-based MoCap | 44970 | 0 | ✗ | ✓ | ✗ | ✗ |
| **Motion-X (Ours)** | 81084 | 144.2 | ✓ | Pseudo GT & MoCap | 81084 | 15.6M | ✓ | ✓ | ✓ | ✓ |

Table 1: Comparisons between *Motion-X* and existing text-motion datasets. The first column shows the name and public year of datasets. *Motion-X* provides both indoor and outdoor whole-body motion and text annotations.

Benchmarks annotated with sequential human motion and text are mainly collected for three tasks: action recognition [33, 27, 34, 28, 35, 36], human object interaction [37, 38, 39, 29, 32, 40], and motion generation [4, 41, 11, 6, 8, 24]. Specifically, KIT Motion-Language Dataset [6] is the first public dataset with human motion and language descriptions, enabling multi-modality motion generation [1, 5]. Although several indoor human motion capture (mocap) datasets have been developed [42, 43, 44, 45], they are scattered. AMASS [11] is noteworthy as it collects and unifies 15 different optical marker-based mocap datasets to build a large-scale motion dataset through a common framework and parameterization via SMPL [46]. This great milestone benefits motion modeling and its downstream tasks. Additionally, BABEL [8] and HumanML3D [4] contribute to the language labels through crowdsourced data collection. BABEL proposes either sequence labels or sub-sequence labels for a sequential motion, while HumanML3D collects three text descriptions for each motion clip from different workers. Thanks to these text-motion datasets, various motion generation methods have rapidly developed and shown advantages in diverse, realistic, and fine-grained motion generation [2, 14, 47, 48, 9, 10].

However, existing text-motion datasets have several limitations, including the absence of facial expressions and hand gestures, insufficient data quantity, limited diversity of motions and scenes, coarse-grained and ambiguous descriptions, and the lack of long sequence motions. To bridge these gaps, we develop a large-scale whole-body expressive motion dataset with comprehensive sequence- and frame-level text labels. We aim to address these limitations and open up new possibilities for future research. We provide quantitative comparisons of *Motion-X* and existing datasets in Tab. 1.

# 3 Motion-X Dataset

## 3.1 Overview

As shown in Tab. 2, we collect *Motion-X* from eight datasets and online videos and provide the following motion and text annotations: 15.6M 3D whole-body SMPL-X annotation, 81.1K sequence-level semantic descriptions (e.g., walking with waving hand and laughing), and frame-level whole-body pose descriptions. Notably, original sub-datasets lack either whole-body motion or text labels and we unify them with our annotation pipeline. All annotations are manually checked to guarantee quality. In Fig. 2, we show the averaged temporal standard deviation of body, hand, and face keypoints of each sub-dataset, highlighting the diversity of hand movements and facial expressions, which fills in the gaps of previous body-only motion data.

| Data | Clip | Frame | GT Motion | P-GT Motion | Text |
|---|---|---|---|---|---|
| AMASS [4] | 26.3K | 5.4M | B | H, F | S |
| HAA500 [27] | 5.2K | 0.3M | - | B, H, F | S |
| AIST [30] | 1.4K | 0.3M | - | B, H, F | S |
| HuMMan [26] | 0.7K | 0.1M | - | B, H, F | S |
| GRAB [29] | 1.3K | 0.4M | B,H | F | S |
| EgoBody [32] | 1.0K | 0.4M | B,H | F | - |
| BAUM [31] | 1.4K | 0.2M | - | F | S |
| IDEA400* | 12.5K | 2.6M | - | B, H, F | - |
| Online Videos* | 32.5K | 6.0M | - | B, H, F | - |
| Motion-X | 81.1K | 15.6M | B, H | B,H,F | S,P |

Table 2: Statistics of sub-datasets. B, H, F are body, hand, and face. S and P are semantic and pose texts. P-GT is pseudo ground truth. * denotes videos are collected by us.

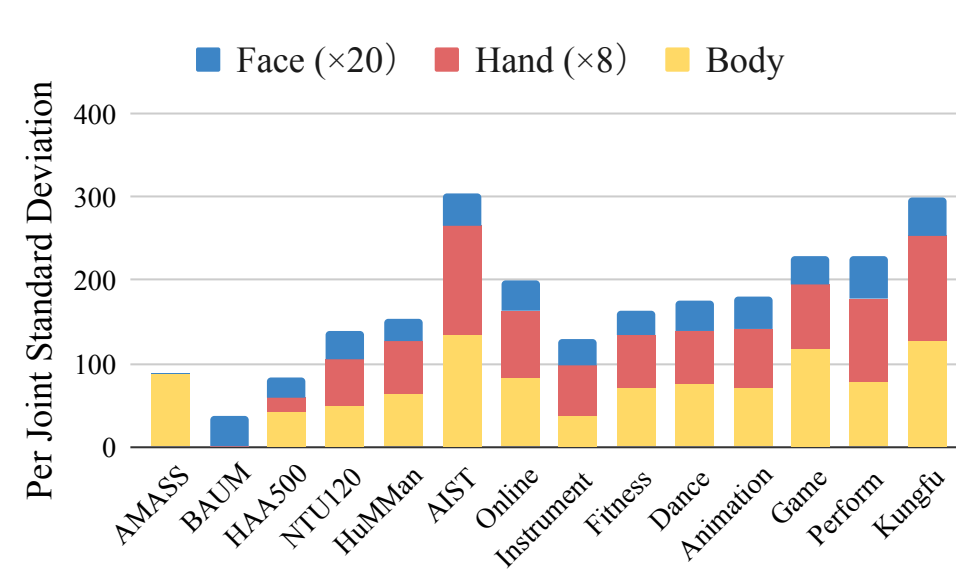

Figure 2: Diversity statistics of the face, hand, and body motions in each subdatasets.

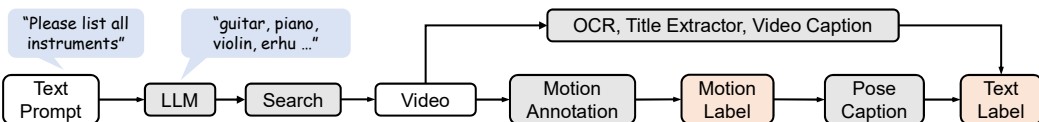

Figure 3: Illustration of the overall data collection and annotation pipeline.

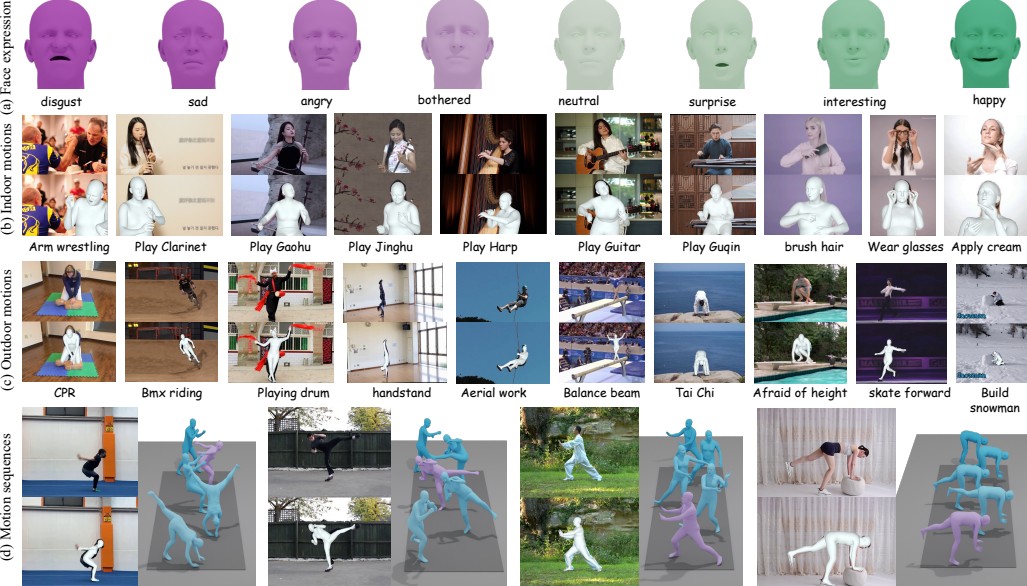

Figure 4: Overview of *Motion-X*. It includes: (a) diverse facial expressions extracted from BAUM [31], (b) indoor motion with expressive face and hand motions, (c) outdoor motion with diverse and challenging poses, and (d) several motion sequences. Purple SMPL-X is the observed frame, and the others are neighboring poses.

## 3.2  Data Collection

As illustrated in Fig. 3, the overall data collection pipeline involves six key steps: 1) designing and sourcing motion text prompts via large language model (LLM) [49], 2) collecting videos, 3) preprocessing candidate videos through human detection and video transition detection, 4) capturing whole-body motion (Sec. 4.1), 5) captioning sequence-level semantic label and frame-level whole-body pose description(Sec. 4.2), and 6) performing the manual inspection.

We gather 37K motion sequences from existing datasets using our proposed unified annotation framework, including the multi-view datasets (AIST [30]), human-scene-interaction datasets (EgoBody [32] and GRAB [29]), single-view action recognition datasets (HAA500 [27], HuMMan [26]), and body-only motion capture dataset (AMASS [11]). For these datasets, steps 1 and 2 are skipped. Notably, only EgoBody and GRAB datasets provide SMPL-X labels with body and hand pose, thus we annotate the SMPL-X label for the other motions. For AMASS, which contains the body and roughly static hand motions, we skip step 4 and fill in the facial expression with a data augmentation mechanism. The facial expressions are collected from a facial datasets BAUM [31] via a face capture and animation model EMOCA [50]. To enrich the expressive whole-body motions, we record an dataset IDEA400, which provides 13K motion sequences covering 400 diverse actions. Details about the processing of each sub-dataset and IDEA400 are available in the appendix.

To improve the appearance and motion diversity, we collect 32.5K monocular videos from online sources, covering various real-life scenes as depicted in Fig. 4. Since human motions and actions are context-dependent and vary with the scenario, we design action categories as motion prompts based on the scenario and function of the action via LLM. To ensure comprehensive coverage of human actions, our dataset includes both general and domain-specific scenes. The general scenes encompass daily actions (e.g., brushing hair, wearing glasses, and applying creams), sports activities (e.g., high knee, kick legs, push-ups), various musical instrument playing, and outdoor scenes (e.g., BMX riding, CPR, building snowman). The inclusion of general scenes helps bridge the gap between existing data and real-life scenarios. In addition, we incorporate domain-specific scenes that require high professional skills, such as dance, Kung Fu, Tai Chi, performing arts, Olympic events, entertainment

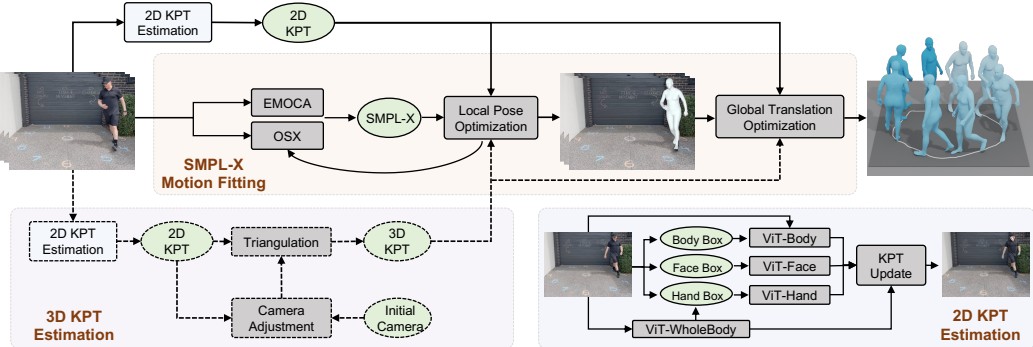

Figure 5: The automatic pipeline for the whole-body motion capture from massive videos, including 2D and 3D whole-body keypoints estimation, local pose optimization, and global translation optimization. This pipeline supports both single- and multi-view inputs. Dashed lines represent the handling of multi-view data exclusively.

shows, games, and animation motions. Based on the prompts describing the above scenes, we run the collection pipeline to gather the data from online sources for our dataset.

# 4 Automatic Annotation Pipeline

## 4.1 Universal Whole-body Motion Annotation

**Overview.** To efficiently capture a large volume of potential motions from massive videos, we propose an annotation pipeline for high-quality whole-body motion capture with *three novel techniques*: (i) hierarchical whole-body keypoint estimation; (ii) score-guided adaptive temporal smoothing for jitter motion refinement; and (iii) learning-based 3D human model fitting for accurate motion capture.

**2D Keypoint Estimation.** 2D Whole-body keypoint estimation poses a challenge due to the small size of the hands and face regions. Although recent approaches have utilized separate networks to decode features of different body parts [51, 52], they often struggle with hand-missing detection and are prone to errors due to occlusion or interaction. To overcome these limitations, we customize a novel hierarchical keypoint annotation method, depicted in the blue box of Fig. 5. We train a ViT-WholeBody based on a ViT-based model [18] on the COCO-Wholebody dataset [51] to estimate initial whole-body keypoints $\mathbf{K}^{2D} \in \mathbb{R}^{133 \times 2}$ with confidence scores. Leveraging the ViT model's ability to model semantic relations between full-body parts, we enhance hand and face detection robustness even under severe occlusion. Subsequently, we obtain the hand and face bounding boxes based on the keypoints, and refine the boxes using the BodyHands detector [53] through an IoU matching operation. Finally, we feed the cropped body, hand, and face regions into three separately pre-trained ViT networks to estimate body, hand and face keypoints, which are used to update $\mathbf{K}^{2D}$.

**Score-guided Adaptive Smoothing.** To address the jitter resulting from per-frame pose estimation in challenging scenarios such as heavy occlusion, truncation, and motion blur, while preserving motion details, we introduce a novel score-guided adaptive smoothing technique into the traditional Savitzky-Golay filter [54]. The filter is applied to a sequence of 2D keypoints of a motion:

$$\bar{\mathbf{K}}_i^{2D} = \sum_{j=-w}^{w} c_j \mathbf{K}_{i+j}^{2D}, \tag{1}$$

where $\mathbf{K}_i^{2D}$ is the original keypoints of the $i_{\text{th}}$ frame, $\bar{\mathbf{K}}_i^{2D}$ is the smoothed keypoints, $w$ corresponds to half-width of filter window size, and $c_j$ are the filter coefficients. Different from existing smoothing methods with a fixed window size [55, 56, 54], we leverage the confidence scores of the keypoints to adaptively adjust the window size to balance between smoothness and motion details. Using a larger window size for keypoints with lower confidence scores can mitigate the impact of outliers.

**3D Keypoint Annotation.** Precise 3D keypoint can boost the estimation of SMPL-X. We utilize novel information from large-scale pre-trained models. Accordingly, for single-view videos, we adopt a pretrained model [57], which is trained on massive 3D datasets, to estimate precise 3D keypoints. For multi-view videos, we utilize bundle adjustment to calibrate and refine the camera parameters, and

then triangulate the 3D keypoints $\bar{\mathbf{K}}^{3D}$ based on the multi-view 2D keypoints. To enhance stability, we adopt temporal smoothing and enforce 3D bone length constraints during triangulation.

**Local Pose Optimization.** After obtaining the keypoints, we perform local pose optimization to register each frame's whole-body model SMPL-X [25]. Traditional optimization-based methods [58, 25] are often time-consuming and may yield unsatisfactory results as they ignore image clues and motion prior. We propose a progressive learning-based human mesh fitting method to address these limitations. Initially, we predict the SMPL-X parameter $\Theta$ with the SOTA whole-body mesh recovery method OSX [15] and face reconstruction model EMOCA [50]. And then, through iterative optimization of the network parameters, we fit the human model parameters $\hat{\Theta}$ to the target 2D and 3D joint positions by minimizing the following functions, achieving an improved alignment accuracy:

$$L_{\text{joint}} = \|\hat{\mathbf{K}}^{3D} - \bar{\mathbf{K}}^{3D}\|_1 + \|\hat{\mathbf{K}}^{2D} - \bar{\mathbf{K}}^{2D}\|_1 + \|\hat{\Theta} - \Theta\|_1. \tag{2}$$

Here, $\hat{\mathbf{K}}^{3D}$ represents the predicted 3D joint positions obtained by applying a linear regressor to a 3D mesh generated by the SMPL-X model. $\hat{\mathbf{K}}^{2D}$ is derived by performing a perspective projection of the 3D keypoints. The last term of the loss function provides explicit supervision based on the initial parameter, serving as a 3D motion prior. To alleviate potential biophysical artifacts, such as interpenetration and foot skating, we incorporate a set of physical optimization constraints:

$$L = \lambda_{\text{joint}} L_{\text{joint}} + \lambda_{\text{smooth}} L_{\text{smooth}} + \lambda_{\text{pen}} L_{\text{pen}} + \lambda_{\text{phy}} L_{\text{phy}}. \tag{3}$$

Here, $\lambda$ are weighting factors of each loss function and $L_{\text{smooth}}$ is a first-order smoothness term:

$$L_{\text{smooth}} = \sum_t \|\hat{\Theta}_{2:t} - \hat{\Theta}_{1:t-1}\|_1 + \sum_t \|\hat{\mathbf{K}}^{3D}_{2:t} - \hat{\mathbf{K}}^{3D}_{1:t-1}\|_1, \tag{4}$$

where $\hat{\Theta}_i$ and $\hat{\mathbf{K}}^{3D}_i$ represent the SMPL-X parameters and joints of the $i$-th frame, respectively. To alleviate mesh interpenetration, we utilize a collision penalizer [59], denoted as $L_{\text{pen}}$. Additionally, we incorporate the physical loss $L_{\text{phy}}$ based on PhysCap [60] to prevent implausible poses.

**Global Motion Optimization.** To improve the consistency and realism of the estimated global trajectory, we perform a global motion optimization based on GLAMR [19] to simultaneously refine the global motions and camera poses to align with video evidence, such as 2D keypoints:

$$L_g = \lambda_{\text{2D}} L_{\text{2D}} + \lambda_{\text{traj}} L_{\text{traj}} + \lambda_{\text{cam}} L_{\text{cam}} + \lambda_{\text{reg}} L_{\text{reg}}, \tag{5}$$

where $L_{\text{2D}}$ represents the 2D keypoint distance loss, $L_{\text{traj}}$ quantifies the difference between the optimized global trajectory and the trajectory estimated by Kama [61]. $L_{\text{reg}}$ enforces regularization on the global trajectory, and $L_{\text{cam}}$ applies a smoothness constraint on the camera parameters.

**Human Verification.** To ensure quality, we manually checked the annotation by removing the motions that do not align with the video evidence or exhibit obvious biophysical artifacts.

### 4.2 Obtaining Whole-body Motion Descriptions

**Sequence motion labels.** The videos in *Motion-X* were collected from online sources and existing datasets. For action-related datasets [26, 27, 16, 28, 11, 29], we use the action labels as one of the sequence semantic labels. Meanwhile, we input the videos into Video-LLaMA [62] and filter the human action descriptions as supplemental texts. When videos contain semantic subtitles, EasyOCR automatically extracts semantic information. For online videos, we also use the search queries generated from LLM [49] as semantic labels. Videos without available semantic information, such as EgoBody [32], are manually labeled using the VGG Image Annotator (VIA) [63]. For the face database BAUM [31], we use the facial expression labels provided by the original creator.

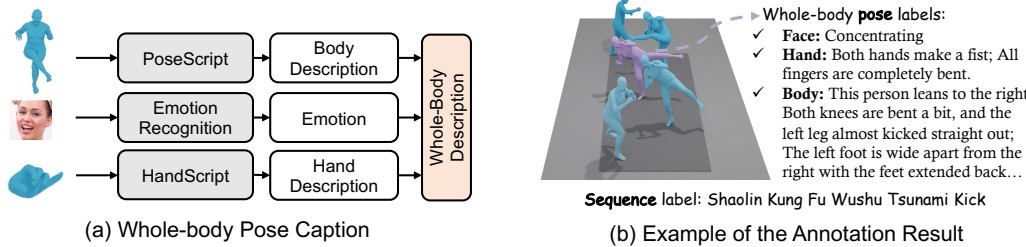

(a) Whole-body Pose Caption

(b) Example of the Annotation Result

Figure 6: Illustration of (a) annotation of the whole-body pose description, and (b) an example of the text labels.

**Whole-body pose descriptions.** The generation of fine-grained pose descriptions for each pose involves three distinct parts: face, body, and hand, as shown in Fig. 6(a). *Facial expression labeling* uses the emotion recognition model EMOCA [50] pretrained on AffectNet [64] to classify the emotion. *Body-specific descriptions* utilizes the captioning process from PoseScript [3], which generates synthetic low-level descriptions in natural language based on given 3D keypoints. The unit of this information is called posecodes, such as *'the knees are completely bent'*. A set of generic rules based on fine-grained categorical relations of the different body parts are used to select and aggregate the low-level pose information. The aggregated posecodes are then used to produce textual descriptions in natural language using linguistic aggregation principles. *Hand gesture descriptions* extends the pre-defined posecodes from body parts to fine-grained hand gestures. We define six elementary finger poses via finger curvature degrees and distances between fingers to generate descriptions, such as *'bent'* and *'spread apart'*. We calculate the angle of each finger joint based on the 3D hand keypoints and determine the corresponding margins. For instance, if the angle between $\vec{V}(\mathbf{K}_{\text{wrist}}, \mathbf{K}_{\text{fingertip}})$ and $\vec{V}(\mathbf{K}_{\text{fingertip}}, \mathbf{K}_{\text{fingeroot}})$ falls between 120 and 160 degrees, the finger posture is labeled as *'slightly bent'*. We show an example of the annotated text labels in Fig. 6(b).

**Summary.** Based on the above annotations, we bulid *Motion-X*, which has 81.1K clips with 15.6M SMPL-X poses and the corresponding pose and semantic text labels.

## 5 Experiment

In this section, we first validate the accuracy of our motion annotation pipeline on the 2D keypoints and 3D SMPL-X datasets. Then, we build a text-driven whole-body motion generation benchmark on *Motion-X*. Finally, we show the effectiveness of *Motion-X* in whole-body human mesh recovery.

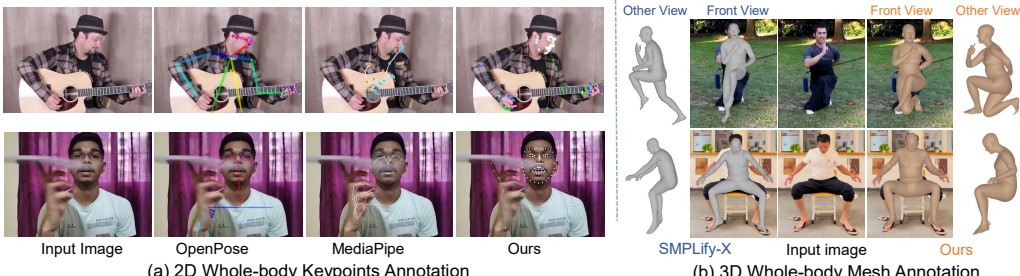

(a) 2D Whole-body Keypoints Annotation  (b) 3D Whole-body Mesh Annotation

Figure 7: Qualitative comparisons of (a) 2D keypoints annotation with widely used methods [65, 66] and (b) the 3D mesh annotation with the popular fitting method [25] with ours.

| Method | hand ↑ AP | hand ↑ AR | face ↑ AP | face ↑ AR | whole-body ↑ AP | whole-body ↑ AR |
|---|---|---|---|---|---|---|
| OpenPose [65] | 38.6 | 43.3 | 76.5 | 84.0 | 44.2 | 52.3 |
| HRNet [67] | 50.3 | 60.2 | 73.7 | 80.9 | 58.2 | 67.1 |
| ViTPose [18] | 47.4 | 59.4 | 59.8 | 70.7 | 57.7 | 69.4 |
| RTMPose-l [68] | 52.3 | 60.0 | 84.4 | 87.6 | 63.2 | 69.4 |
| Ours | **64.9** | **74.0** | **91.6** | **94.4** | **73.5** ↑16.3% | **80.3** ↑15.7% |

| Method | PA-MPJPE ↓ | PA-MPVPE ↓ | MPVPE ↓ |
|---|---|---|---|
| Hand4Whole [17] | 58.9 | 50.3 | 79.2 |
| OSX [15] | 55.6 | 48.7 | 70.8 |
| PyMAF-X [69] | 52.8 | 50.2 | 64.9 |
| SMPLify-X [25] | 62.6 | 52.9 | - |
| Ours | 33.5 | 31.8 | 44.7 ↓30.1% |
| Ours w/GT 3Dkpt | **23.9** | **19.7** | **30.7** ↓52.7% |

(a) Evaluation result on COCO-Wholebody [51] dataset.  (b) Reconstruction error on EHF [25] dataset.

Table 3: Evaluation of motion annotation pipeline on (a) 2D keypoints and (b) 3D SMPL-X datasets.

### 5.1 Evaluation of the Motion Annotation Pipeline

**2D Keypoints Annotation.** We evaluate the proposed 2D keypoint annotation method on the COCO-WholeBody [51] dataset, and compare the evaluation result with four SOTA keypoints estimation methods [65, 67, 18, 68]. We use the same input image size of $256 \times 192$ for all the methods to ensure a fair comparison. From Tab. 3(a), our annotation pipeline significantly surpasses existing methods by over 15% average precision. Additionally, we provide qualitative comparisons in Fig. 7(a), illustrating the robust and superior performance of our method, especially in challenging and occluded scenarios.

**3D SMPL-X Annotation.** We evaluate our learning-based fitting method on the EHF [25] dataset and compare it with four open-sourced human mesh recovery methods. Following previous works, we employ mean per-vertex error (MPVPE), Procrusters aligned mean per-vertex error (PA-MPVPE), and Procrusters aligned mean per-joint error (PA-MPJPE) as evaluation metrics (in mm). Results in Tab. 3(b) demonstrate the superiority of our progressive fitting method (over 30% error reduction). Specifically, PA-MPVPE is only 19.71 mm when using ground-truth 3D keypoints as supervision. Fig. 7(b) shows the annotated mesh from front and side view, indicating reliable 3D SMPL-X annotations with reduced depth ambiguity. More results are presented in Appendix due to page limits.

| Methods | R Precision ↑ | | | FID↓ | MM Dist↓ | Diversity→ | MModality |
| | Top 1 | Top 2 | Top 3 | | | | |
|---|---|---|---|---|---|---|---|
| Real | $0.573^{\pm.005}$ | $0.765^{\pm.003}$ | $0.850^{\pm.005}$ | $0.001^{\pm.001}$ | $2.476^{\pm.002}$ | $13.174^{\pm.227}$ | - |
| MDM [14] | $0.290^{\pm.011}$ | $0.459^{\pm.010}$ | $0.577^{\pm.008}$ | $2.094^{\pm.230}$ | $6.221^{\pm.115}$ | $11.895^{\pm.354}$ | $2.624^{\pm.083}$ |
| MLD [2] | $0.440^{\pm.002}$ | $0.624^{\pm.004}$ | $0.733^{\pm.003}$ | $0.914^{\pm.056}$ | $3.407^{\pm.020}$ | $13.001^{\pm.245}$ | $2.558^{\pm.084}$ |
| T2M-GPT [48] | $0.502^{\pm.004}$ | $0.697^{\pm.005}$ | $0.791^{\pm.007}$ | $0.699^{\pm.012}$ | $3.192^{\pm.035}$ | $\mathbf{13.132}^{\pm.127}$ | $2.510^{\pm.027}$ |
| MotionDiffuse [9] | $\mathbf{0.559}^{\pm.001}$ | $\mathbf{0.748}^{\pm.004}$ | $\mathbf{0.842}^{\pm.003}$ | $\mathbf{0.457}^{\pm.007}$ | $\mathbf{2.542}^{\pm.018}$ | $13.576^{\pm.161}$ | $1.620^{\pm.152}$ |

Table 4: Benchmark of text-driven motion generation on *Motion-X* test set. '→' means results are better if the metric is closer to the real motions and $\pm$ indicates the 95% confidence interval. The best results are in **bold**.

| Train Set | HumanML3D (Test) | | | | Motion-X (Test) | | | |
| | R-Precision↑ | FID↓ | Diversity→ | MModality | R-Precision↑ | FID↓ | Diversity→ | MModality |
|---|---|---|---|---|---|---|---|---|
| Real (GT) | $0.749^{\pm.002}$ | $0.002^{\pm.001}$ | $9.837^{\pm.084}$ | - | $0.850^{\pm.005}$ | $0.001^{\pm.001}$ | $13.174^{\pm.227}$ | - |
| HumanML3D | $0.657^{\pm.004}$ | $1.579^{\pm.050}$ | $10.098^{\pm.052}$ | $2.701^{\pm.143}$ | $0.570^{\pm..003}$ | $12.309^{\pm.127}$ | $9.529^{\pm.165}$ | $2.960^{\pm.066}$ |
| Motion-X | $\mathbf{0.695}^{\pm.005}$ | $\mathbf{0.999}^{\pm.042}$ | $\mathbf{9.871}^{\pm.099}$ | $\mathbf{2.827}^{\pm.138}$ | $\mathbf{0.733}^{\pm.003}$ | $\mathbf{0.914}^{\pm.056}$ | $\mathbf{13.001}^{\pm.245}$ | $2.558^{\pm.084}$ |

Table 5: Cross-dataset comparisons of HumanML3D and *Motion-X*. We train MLD on the training set of HumanML3D and *Motion-X*, respectively, then evaluate it on their test sets.

## 5.2 Impact on Text-driven Whole-body Motion Generation

**Experiment Setup.** We randomly split *Motion-X* into the train (80%), val (5%), and test (15%) sets. SMPL-X is adopted as the motion representation for expressive motion generation.

**Evaluation metrics.** We adopt the same evaluation metrics as [4], including Frechet Inception Distance (FID), Multimodality, Diversity, R-Precision, and Multimodal Distance. Due to the page limit, we leave more details about experimental setups and evaluation metrics in the appendix.

**Benchmarking Motion-X.** We train and evaluate four diffusion-based motion generation methods, including MDM [14], MLD [2], MotionDiffuse [9] and T2M-GPT [48] on our dataset. Since previous datasets only have sequence-level motion descriptions, we keep similar settings for minimal model adaptation and take the semantic label as text input. The evaluation is conducted with 20 runs (except for Multimodality with 5 runs) under a 95% confidence interval. From Tab. 4, MotionDiffuse demonstrates a superior performance across most metrics. However, it scores the lowest in Multimodality, indicating that it generates less varied motion. Notably, T2M-GPT achieves comparable performance on our dataset while maintaining high diversity, indicating our large-scale dataset's promising prospects to enhance the GPT-based method's efficacy. MDM gets the highest Multimodality score with the lowest precision, indicating the generation of noisy and jittery motions. The highest Top-1 precision is 55.9%, showing the challenges of *Motion-X*. MLD adopts the latent space design, making it fast while maintaining competent results. Therefore, we use MLD to conduct the following experiments to compare *Motion-X* with the existing largest motion dataset HumanML3D and ablation studies.

**Comparison with HumanML3D.** To validate the richness, expressiveness, and effectiveness of our dataset, we conduct a comparative analysis between *Motion-X* and HumanML3D, which is the largest existing dataset with text-motion labels. We replace the original vector-format poses of HumanML3D with the corresponding SMPL-X parameters from AMASS [11], and randomly extract facial expressions from BAUM [31] to fill in the face parameters. We train MLD separately on the training sets of *Motion-X* and HumanML3D, then evaluate both models on the two test sets. The results in Tab. 5 reveal some valuable insights. Firstly, *Motion-X* exhibits greater diversity (**13.174**) than HumanML3D (**9.837**), as evidenced by the real (GT) row. This indicates a wider range of motion types captured by *Motion-X*. Secondly, the model pretrained on *Motion-X* and then fine-tuned on HumanML3D subset performs well on

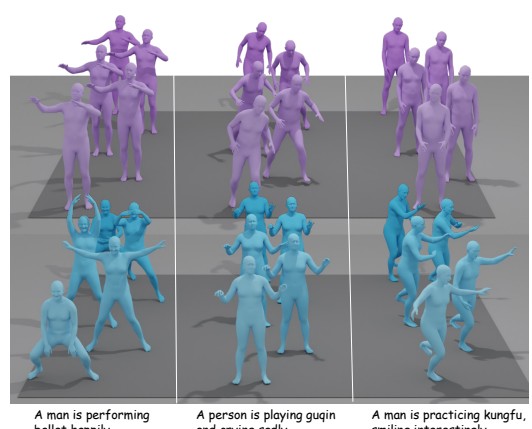

A man is performing ballet happily.    A person is playing guqin and crying sadly.    A man is practicing kungfu, smiling interestingly.

Figure 8: Visual comparisons of motions generated by MLD [2] trained on HumanML3D (in purple) or *Motion-X* (in blue). Please zoom in for a detailed comparison. The model trained with *Motion-X* can generate more accurate and semantic-corresponded motions.

| Semantic Label | Pose Discription | | | FID↓ |
|:---:|:---:|:---:|:---:|:---:|
| | face text | body text | hand text | |
| ✓ | | | | $0.914^{\pm.056}$ |
| ✓ | ✓ | | | $0.784^{\pm.032}$ |
| ✓ | ✓ | ✓ | | $0.671^{\pm.016}$ |
| ✓ | ✓ | ✓ | ✓ | $\mathbf{0.565^{\pm.036}}$ |

Table 6: Ablation study of text inputs.

| Method | EHF [25] ↓ | | | AGORA [70] ↓ | | |
|:---:|:---:|:---:|:---:|:---:|:---:|:---:|
| | all | hand | face | all | hand | face |
| w/o Motion-X | 79.2 | 43.2 | 25.0 | 185.6 | 73.7 | 82.0 |
| w/ Motion-X | **73.0** | **41.0** | **22.6** | **184.1** | **73.3** | **81.4** |

Table 7: Mesh recovery errors of Hand4Whole [17] using different training datasets. MPVPE (mm) is reported.

the HumanML3D test set, even better than the intra-data training. These superior performances stem from the fact that *Motion-X* encompasses diverse motion types from massive outdoor and indoor scenes. For a more intuitive comparison, we provide the visual results of the generated motion in Fig. 8, where we can clearly see that the model trained on *Motion-X* excels at synthesizing semantically corresponding motions given text inputs. These results prove the significant advantages of *Motion-X* in enhancing expressive, diverse, and natural motion generation.

**Ablation study of text labels.** In addition to sequence-level semantic labels, the text labels in *Motion-X* also include frame-level pose descriptions, which is an important characteristic of our dataset. To assess the effectiveness of pose description, we conducted an ablation study on the text labels. The baseline model solely utilizes the semantic label as the text input. Since there is no method to use these labels, we simply sample a single sentence from the pose descriptions randomly, concatenate it with the semantic label, and feed the combined input into the CLIP text encoder. Interestingly, from Tab. 6, adding additional face and body pose texts brings consistent improvements, and combining whole-body pose descriptions results in a noteworthy $38\%$ reduction in FID. These results validate that the proposed whole-body pose description contributes to generating more accurate and realistic human motions. More effective methods to utilize these labels can be explored in the future.

## 5.3 Impact on Whole-body Human Mesh Recovery

As discovered in this benchmark [21], the performance of mesh recovery methods can be significantly improved by utilizing high-quality pseudo-SMPL labels. *Motion-X* provides a large volume of RGB images and well-annotated SMPL-X labels. To verify its usefulness in the 3D whole-body mesh recovery task, we take Hand4Whole [17] as an example and evaluate MPVPE on the widely-used AGORA val [71] and EHF [25] datasets. For the baseline model, we train it on the commonly used COCO [51], Human3.6M [72], and MPII [73] datasets. We then train another model by incorporating an additional $10\%$ of the single-view data sampled from *Motion-X* while keeping the other setting the same. As shown in Tab. 7, the model trained with *Motion-X* shows a significant decrease of $7.8\%$ in MPVPE on EHF and AGORA compared to the baseline model. The gains come from the increase in diverse appearances and poses in *Motion-X*, indicating the effectiveness and accuracy of the motion annotations in *Motion-X* and its ability to benefit the 3D reconstruction task.

## 6 Conclusion

In this paper, we present *Motion-X*, a comprehensive and large-scale 3D expressive whole-body human motion dataset. It addresses the limitations of existing mocap datasets, which primarily focus on indoor body-only motions with limited action types. The dataset consists of 144.2 hours of whole-body motions and corresponding text labels. To build the dataset, we develop a systematic annotation pipeline to annotate 81.1K 3D whole-body motions, sequence-level motion semantic labels, and 15.6M frame-level whole-body pose descriptions. Comprehensive experiments demonstrate the accuracy of the motion annotation pipeline and the significant benefit of *Motion-X* in enhancing expressive, diverse, and natural motion generation, as well as 3D whole-body human mesh recovery.

**Limitation and future work.** There are two main limitations of our work. Firstly, the motion quality of our markless motion annotation pipeline is inevitably inferior to the multi-view marker-based motion capture system. Secondly, during our experiment, we found out that existing evaluation metrics are not always consistent with visual results. Thus, there is a need for further development of the motion generation models and evaluation metrics. As a large-scale dataset with multiple modalities, e.g., motion, text, video, and audio, *Motion-X* holds great potential for advancing downstream tasks, such as motion prior learning, understanding, and multi-modality pre-training. Besides, our large-scale dataset and scalable annotation pipeline open up possibilities for combining this task with a large language model (LLM) to achieve an exciting motion generation result in the future. With *Motion-X*, we hope to benefit and facilitate further research in relevant fields.

## Acknowledgements

This research is partially funded through National Key Research and Development Program of China (Project No. 2022YFB36066), in part by the Shenzhen Science and Technology Project under Grant (JCYJ20220818101001004, JSGG20210802153150005), National Natural Science Foundation of China (62271414), the Young Scientists Fund of the National Natural Science Foundation of China under grant No.62106154, the Natural Science Foundation of Guangdong Province, China (General Program) under grant No.2022A1515011524, and Shenzhen Science and Technology Program JCYJ20220818103001002 and Shenzhen Science and Technology Program ZDSYS20211021111415025.

Thanks to everyone who participated in shooting the IDEA400 dataset and discussions. Thanks to Linghao Chen for helping to design the icons and website pages.

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
