# OpenReview forum: "Motion-X: A Large-scale 3D Expressive Whole-body Human Motion Dataset"
_NeurIPS.cc/2023/Track/Datasets_and_Benchmarks — NeurIPS 2023 Datasets and Benchmarks Poster_

### Official Review · Reviewer_4k8C · 2023-07-06

**Rating:** 6
**Confidence:** 5
**Correctness:** Yes

**Strengths:**

As the authors mentioned, most existing datasets have body-only annotations. Hence, the newly introduced dataset will be greatly useful as it additionally provides 3D hand poses, 3D facial expression, and text descriptions.


**Additional Feedback:**

No

**Clarity:**

There are several things to be clarified.

1. The numbers of Table 1 and 2 are different. For example, the number of clips of AMASS in Table 1 is 11K, but in Table 2, the number is 26K. Why are they different?

2. Does L167 represent EFT [B]? Or optimize SMPL-X parameters like SMPLify-X?
[B] Joo, Hanbyul, Natalia Neverova, and Andrea Vedaldi. "Exemplar fine-tuning for 3d human model fitting towards in-the-wild 3d human pose estimation." 2021 International Conference on 3D Vision (3DV). IEEE, 2021.

3. During the manual verification of L184, did annotators also rotate the meshes to check depth ambiguity?

4. In Table 3 (a), what is ours? Is it ‘2D KPT Estimation’ in Fig. 5? What makes ours in Table 3 (a) much better than previous works? L215 says annotation, but all other methods in Table 3 (a) are not for the annotation.

5. In Table 3 (b), please clarify how the authors achieve their numbers. The authors used Fig. 5? L221 says annotation, but all other methods in Table 3 (b) are not for the annotation. According to Fig. 5, the only difference from OSX for the EHF evaluation is the additional local pose optimization, where the target 2D pose comes from ‘2D KPT Estimation’. But the numbers between OSX and the proposed one in Table 3 (b) are quite different. Does that mean the local pose optimization is greatly important?


**Documentation:**

Yes

**Limitations:**

Yes

**Opportunities For Improvement:**

1. Quality of mocap datasets in Motion-X.
Among mocap datasets, only AMASS, GRAB, and EgoBody provide both body and hand poses (L111), and AMASS has the largest scale among them. However, AMASS does not have accurate 3D hand poses and actually, most of the hand poses are just a neutral pose. In this sense, using the 3D hand pose annotation from AMASS does not make the AMASS part of the Motion-X a ‘whole-body’ dataset. In addition, facial expressions are randomly brought from another dataset (L114), which can make body motion and facial expression inconsistent.

2. Quality of online video data in Motion-X.
Fig. 3 and Fig. 5 show how the authors annotate 3D whole-body pose from online videos. One critical limitation of this annotation pipeline is inherent depth, scale, and occlusion ambiguity. Fig. 4 shows various examples of 3D whole-body annotation from online videos. I’m pretty sure that 3D whole-body annotations from ‘brush hair’, ‘Wear glasses’, and ‘Apply cream’ are wrong when seeing from the side view by rotating them due to the depth ambiguity. 3D hands should actually almost touch (or make contact) the face, but due to the depth ambiguity, 3D hands are often far from the face although it looks reasonable from the front view. In this sense, the annotations from online videos should be called pseudo-GT, which fortunately the authors mentioned in L285. But in many descriptions, such as Table 1 and 2, the authors compared Motion-X with datasets that have GT (not pseudo-GT). This comparison does not make sense as they have different characteristics: datasets with GT do not have corresponding in-the-wild images, while datasets with pseudo-GT (like Motion-X) have. Hence, the authors should exclude all data from online videos in Table 1 and 2 and explicitly say that annotations from the online videos are pseudo-GT.

3. Lack of comparison to approaches for obtaining pseudo-GT.
Recently, several methods are proposed to obtain pseudo-GT from in-the-wild images, such as images from MSCOCO. Among them, NeuralAnnot [A] introduces 3D pseudo-GT of MSCOCO. The authors should compare their annotation pipeline with NeuralAnnot. Basically, NeuralAnnot trains a 3D whole-body pose estimator on the training set of MSCOCO and tests on the training set, where the testing outputs become 3D pseudo-GT. As NeuralAnnot is a general framework and not restricted with network architectures, training OSX on the training set of MSCOCO and testing it on the training set could be a way to use NeuralAnnot.

[A] Moon, Gyeongsik, Hongsuk Choi, and Kyoung Mu Lee. "NeuralAnnot: Neural annotator for 3D human mesh training sets." CVPRW. 2022.


**Relation To Prior Work:**

Yes

**Summary And Contributions:**

This paper presents a large-scale 3D human pose dataset, which contains 3D whole-body annotations, including 3D poses of body and hands and facial expression. As most of existing large-scale datasets provide only body-only annotations, the newly introduced dataset, Motion-X, could be highly useful. In addition, the authors included text description of 3D whole-body motion, which enables multimodal learning as well. For the annotation, the authors introduced an automatic annotation pipeline.

---

> ### Author Response · Authors · 2023-08-21
> **Rebuttal by Authors - Part 1**
>
> &nbsp;
> ### Response to Reviewer 4k8C - Part 1
>
> &nbsp;
>
> Thanks for your valuable comments.
>
> &nbsp;
>
> `Q-1`: Quality of motion capture datasets.
>
> `A-1`: (a) Hand motion. Yes, AMASS predominantly comprises neutral hand poses and does not provide GT hand poses. We have clarified it in Table 2 of the revised paper. Nevertheless, we included it in Motion-X due to its high-quality body motion, which enhances body-part motion generation. In our motion generation experiments, we intentionally exclude hand-part motion loss calculations from AMASS to mitigate potential negative effects stemming from inaccurate hand poses.
>
> (b) Facial expression. As shown in Figure 2 of the Appendix, body motions often decouple from facial expressions. For example, both "sitting happily" and "sitting sadly" are plausible motions. Thus, we fill in the facial motion of AMASS and GRAB with BAUM and incorporate relevant keywords in the semantic description, which can ensure uniformity across sub-datasets while maintaining the semantic consistency between text and motion.  But as you mentioned, this may introduce inconsistency. We would consider addressing this by manual selection or motion style transferring.
>
> &nbsp;
>
> `Q-2`: Quality of online video data. Clarification of pseudo-GT.
>
> `A-2`: (a) We agree that it is indeed challenging to annotate high-quality motion from monocular videos, especially for the depth ambiguity issue. We've dedicated considerable efforts to enhance motion quality by improving input video quality and our annotation pipeline: (i) We've developed preprocessing techniques to curate high-quality videos featuring subjects with high resolution and full body visibility, while effectively excluding severely truncated or occluded instances, as detailed in Sec. G.1 of the revised Appendix. (ii) We are keeping improving our annotation pipeline. Recent advancements introduced depth information using a SOTA method [1] and the use of physical constraints or optimization. These approaches leverage learned depth priors, effectively addressing depth ambiguity. Additional visualization results can be found in Fig. 4 and Fig. 5 of the Appendix and on our website.
>
> (b) Thanks for your kind reminder. The proposed annotations of the online videos are pseudo-GT, and we have specified them in Table 1 and Table 2 of the revised paper.
>
> &nbsp;
>
> `Q-3`: Comparison with NeuralAnnot.
>
> `A-3`: Thanks for your kind suggestion. However, NeuralAnnot is not open-sourced, and it only provides the annotated data, so we compare the widely-used open-sourced annotation method SMPLify-X in Table 3. When annotating 2D in-the-wild videos, our method has many advantages over NeuralAnnot in:
>
> (a) Estimation of global motion. NeuralAnnot predicts local motion in virtual camera coordinates, while we predict global motion in world coordinates.
>
> (b) Loss function. During local pose optimization, NeuralAnnot utilizes the 2D joint loss and L2 regularization. Our approach employs 2D and 3D joint loss, temporal smoothing, interpenetration, and physical loss, offering more accurate, smoothed, and consistent annotations.
>
> (c) 3D pose prior. NeuralAnnot employs VPoser for pose prior and predicts the latent code of VPoser during fitting. In contrast, we utilize the implicit pose prior to OSX and EMOCA and directly predict the 3D joint rotations during fitting. This also leads to significantly different iterative mechanisms between NeuralAnnot and our method.
>
> Here, we compare our method with NeuralAnnot by computing:
>
> (a) Direct 3D annotation errors on a well-known test set EHF. We train and test OSX on EHF using NeuralAnnot's loss function. As shown, our method achieves notably lower MPVPE.
>
> |             | PA-MPJPE | PA-MPVPE | MPVPE |
> | :---------: | :------: | :------: | :---: |
> | NeuralAnnot |   44.5   |   46.8   | 67.8  |
> |    Ours     |   33.5   |   31.8   | 44.7  |
>
> Table R4: Comparison of the direct 3D annotation error on the EHF dataset between our method and NeuralAnnot.
>
> (b) Indirect 3D annotation error. MSCOCO lacks 3D GTs. NeuralAnnot provided the corresponding pseudo-GTs, which have been widely used in many mesh recovery model training. We also annotate the SMPL-X on MSCOCO as another pseudo-GTs. We indirectly measure the quality of the two pseudo-GTs by comparing how much the pseudo-GTs are beneficial for the training of the Hand4Whole model. As shown, training with our MSCOCO pseudo-GTs can achieve significantly lower SMPL-X estimation error on the EHF dataset.
>
> |             | PA-MPJPE | PA-MPVPE | MPVPE |
> | :---------: | :------: | :------: | :---: |
> | NeuralAnnot |   51.7   |   53.1   | 79.2  |
> |    Ours     |   49.3   |   50.8   | 76.4  |
>
> Table R5: Comparison of the indirect 3D annotation error on the EHF dataset between our method and NeuralAnnot.
>
> &nbsp;
>
>
> Reference
>
> [1] Zoedepth: Zero-shot transfer by combining relative and metric depth. Arxiv.

---

> > ### Author Response · Authors · 2023-08-21
> > **Rebuttal by Authors - Part 2**
> >
> > &nbsp;
> >
> > ### Response to Reviewer 4k8C - Part 2
> >
> > &nbsp;
> >
> > `Q-4`: Clarification of AMASS clip number.
> >
> > `A-4`: In Table 2, we split the long motion sequences into several short clips. Thus, the sequence number of AMASS in Table 2 is larger than that in Table 1. Sorry for the confusion. We have clarified it in Sec. B.2 in the revised Appendix.
> >
> > &nbsp;
> >
> > `Q-5`: Clarification of L167.
> >
> > `A-5`: As mentioned in L167, we optimize the network parameters instead of SMPL-X parameters. Thus, it's more relevant to EFT instead of SMPlify-X. Notably, our method also differs from EFT in many aspects, such as the loss functions, iterative mechanism, pose prior, etc.
> >
> > &nbsp;
> >
> > `Q-6`: Clarification of manual verification.
> >
> > `A-6`: Yes, we check the depth ambiguity by visualizing the global motion in the world coordinate with a side view, like Figure 8 and videos on our website. Specifically, we manually verify the validity of the motion sequences through the following steps:
> >
> > (a) Project the body mesh onto the image plane, overlaying it on the original RGB image for 2D alignment inspection.
> >
> > (b) Utilize Blender to visualize the human motion in world coordinates, examining global trajectory plausibility and continuity.
> >
> > Finally, motion sequences exhibiting coherent global trajectories, and image alignment are labeled as valid, while others are deemed invalid.
> >
> > &nbsp;
> >
> > `Q-7`: Clarification of Table 3(a).
> >
> > `A-7`: Yes, "ours" in Table 3(a) refers to the proposed hierarchical keypoint estimation method described in Lines 134 - 145 of our paper. The advantages of our method mainly stem from:
> >
> > (a) Utilizing strong backbones. We trained four large-scale ViT networks with extensive datasets for whole-body, body, hand, and face keypoints detection, respectively. These experts take the upsampled cropped part-specific image as input and can estimate accurate part-specific keypoints separately. In contrast, other methods (e.g., OpenPose) either use a lightweight backbone lacking powerful representation ability, or use one network (e.g., ViTPose-Wholebody) for whole-body keypoints estimation, suffering from hand and face resolution issues.
> >
> > (b) Combining keypoints with SOTA hand detector. To solve the hand omission issue, which is prevalent in prior methods like OpenPose, we combine the whole-body keypoints with the SOTA hand detector and greatly improve the robustness under occlusion.
> >
> > "Annotation" here refers to automated methods estimating pseudo-GTs for downstream tasks, not just manual annotation. For example, one of our competitors in Table 3(a), OpenPose, is widely used for keypoint annotation to create several datasets, e.g., TalkShow [2], AIST [3], HuMMan [4].
> >
> > &nbsp;
> >
> > `Q-8`: Clarification of Table 3(b).
> >
> > `A-8`: Yes, we annotate the 3D mesh of EHF by estimating 2D and 3D keypoints, then refining through local pose optimization. The 2D keypoints offer precise 2D clues for alignment, while the 3D keypoints estimated by SOTA method [5] can provide crucial 3D body prior learned from dozens of 3D datasets, alleviating depth ambiguity. Moreover, the elaborately designed iterative manner and loss functions are also critical for the overall annotation accuracy. For example, The temporal term in Eq. (4) leverages temporal info for better, smoother outcomes. The local pose optimization is indeed pivotal, synergizing the informative 2D and 3D keypoints iteratively with well-crafted regularization.
> >
> > &nbsp;
> >
> > Reference
> >
> > [1] Zoedepth: Zero-shot transfer by combining relative and metric depth. Arxiv.
> >
> > [2] Generating Holistic 3D Human Motion from Speech. In CVPR 2023.
> >
> > [3] AI Choreographer: Music Conditioned 3D Dance Generation with AIST++.  ICCV 2021.
> >
> > [4] HuMMan: Multi-Modal 4D Human Dataset for Versatile Sensing and Modeling. ECCV 2022.
> >
> > [5] Learning 3D Human Pose Estimation from Dozens of Datasets using a Geometry-Aware Autoencoder to Bridge Between Skeleton Formats. WACV 2023.

---

> > > ### Comment · Reviewer_4k8C · 2023-08-23
> > >
> > > Thanks for the clarifications.
> > > Most of my concerns are addressed.
> > > I'd like to suggest to include all the above answers in the final version.

---

> > > > ### Author Response · Authors · 2023-08-23
> > > >
> > > > Dear reviewer 4k8C:
> > > >
> > > > Thanks for your reply and suggestions. We will include the answers in the revised final version later.
> > > >
> > > > Best regards,
> > > > Authors

---

### Official Review · Reviewer_Kpcx · 2023-07-20
**A large-scale 3D human motion dataset with rich multimodal annotation**

**Rating:** 6
**Confidence:** 4
**Correctness:** Yes
**Clarity:** Yes

**Strengths:**

1. Human motion modeling has been an interesting research topic. The proposed dataset combines multiple existing datasets on human motion, and augments them with consistent annotation. It can benefit the research community on human-motion-related topics.
2. An automatic pipeline is proposed for the data processing and annotation purpose. Experimental results show that the proposed pipeline outperforms other methods on 2D keypoint and 3D SMPL-X annotation.
3. The authors show by experiments that the proposed dataset can help data-drive methods on body motion generation and human mesh recovery tasks.


**Additional Feedback:**

1. How is the manual inspection done? Is it done for every single annotation?
2. Does every data sample come with video and audio? For example, AMASS and GRAB do not provide RGB videos, are there synthetic videos generated for them?

**Documentation:**

Yes

**Ethics:**

This paper collected massive online videos from multiple sources, so there could be copyright or privacy concerns.

**Limitations:**

Limitations are well discussed.

**Opportunities For Improvement:**

1. Since this dataset is so complex, it is important to have ablation study on the dataset itself. The data are collect from multiple sources, and each subset might be annotated in different ways, resulting in potential inconsistent quality. For example, the body poses from motion capture dataset are trustworthy, while monocular pose annotation may not be reliable. For text descriptions, some are generated by LLM, while others are manually annotated. It is not clear if different annotations can be treated in the same way. It would be good to see the difference on some tasks between using the full dataset and using only high-quality subset. In the data release, the authors may need to clearly state the annotation method for each data sample, so people can decide whether they want to mix them or not.
2. The body motion annotation for online videos are not satisfactory. Many of the videos show obvious jittering and unreal feet sliding. Although some temporal smoothing is applied, and the authors acknowledge this limitation, it would be better if this can be further improved.


**Relation To Prior Work:**

Yes

**Summary And Contributions:**

This paper constructs a large-scale human dataset annotated with body, face, and hand motions (as sequence of SMPL-X parameters), as well as sequence-level and frame-level text descriptions. Part of the data are associated with video and audio. Considering the multimodality nature of this dataset, it can benefit the study of many human-motion-related topics.

---

> ### Author Response · Authors · 2023-08-21
> **Rebuttal by Authors**
>
> &nbsp;
> ### Response to Reviewer Kpcx
>
> &nbsp;
>
> Thanks for your valuable comments.
>
> &nbsp;
>
> `Q-1`: Ablation study on subsets and clarification of annotation methods for each sub-dataset.
>
> `A-1`: Thanks for your thoughtful question.
>
> (a) Our paper focuses on showcasing how a large-scale dataset, annotated through a scalable pipeline, enhances whole-body motion generation and mesh recovery. To ensure experiment effectiveness, we employ the entire dataset rather than specific subsets.  Here, we conduct a detailed ablation study of sub-datasets in whole-body motion generation task, categorizing Motion-X into three groups based on motion quality: GT body-only motions (AMASS), GT body and hand motions (EgoBody, GRAB), and pseudo-GT whole-body motions (other subsets), respectively. We train MLD with different groups and evaluate on Motion-X testset. This study affirms the positive impact of all sub-datasets on whole-body motion generation.
>
> |  AMASS   | GRAB+EgoBody |  Others  |  FID  |
> | :------: | :----------: | :------: | :---: |
> | &#10004; |              |          | 12.31 |
> | &#10004; |   &#10004;   |          | 5.60  |
> | &#10004; |              | &#10004; | 1.63  |
> | &#10004; |   &#10004;   | &#10004; | 0.91  |
>
> Table R3: Ablation study of different subsets on whole-body motion generation task. We train MLD on different subsets and evaluate on Motion-X test set.
>
> (b) We have introduced the processing and annotation of each sub-dataset in Sec. B.2 of Appendix. This information will be emphasized upon the dataset's release.
>
> &nbsp;
>
> `Q-2`: Improve the motion quality of online videos.
>
> `A-2`: Thanks for your advice. It's indeed very challenging to annotate perfect motion from monocular videos. We're constantly enhancing our annotation pipeline, improving motion quality with reduced jitter and foot sliding in the newest version. Check our website (https://motion-x-dataset.github.io) for more visualization results. While not perfect, pseudo-GT motions prove valuable for generation and recovery tasks, as shown in our experiments and previous work [1, 2].
>
> &nbsp;
>
> `Q-3`: Ethical concern about online videos.
>
> `A-3`: In fact, a large portion of our videos are either authorized by original owners or collected from public platforms like Pixabay that permit commercial and academic use.  However, since our dataset primarily focuses on the text-driven motion generation task, we would only provide the annotated text-motion labels and would not distribute the RGB videos. If RGB videos are needed, we will follow previous works of YouTube-8M [3] and LAION [4] to provide the links to the online videos for the users to download by themselves.
>
> &nbsp;
>
> `Q-4`: Question about manual inspection.
>
> `A-4`: As shown in Fig. 5, our motion annotation pipeline can be divided into three steps, including keypoints estimation, local pose optimization, and global translation optimization. We identified common attributes among failed motion sequences, such as excessive keypoint jitter (evaluated by keypoint temporal deviation), or poor alignment (evaluated by 2D/3D keypoints fitting error). Thus, after annotation, we calculate the temporal deviation and keypoint fitting error of the motion sequences for preliminary screening, and verify the validity of the motion sequences with notable jitter or error through the following steps:
>
> (a) Project the body mesh onto the image plane, overlaying it on the original RGB image for 2D alignment inspection.
>
> (b) Utilize Blender to visualize the human motion in world coordinates, examining global trajectory plausibility and continuity.
>
> Motion sequences exhibiting coherent global trajectories and image alignment are labeled as valid, while others are deemed invalid.
>
> Notably, we validate through visualization that the automatic filtering using temporal deviation or projection error exhibits a high level of consistency with human assessment, highlighting the effectiveness and efficiency of this automated preliminary screening process.
>
> &nbsp;
>
> `Q-5`: Does every data sample come with video and audio?
>
> `A-5`: No, as indicated in Table 1 of the Appendix, AMASS and GRAB do not provide RGB videos and audio, whereas the other sub-datasets offer.
>
> &nbsp;
>
> Reference
>
> [1] Text-Conditional Contextualized Avatars For Zero-Shot Personalization. Arxiv.
>
> [2] Benchmarking and Analyzing 3D Human Pose and Shape Estimation Beyond Algorithms. NeurIPS 2022.
>
> [3] YouTube-8M: A Large-Scale Video Classification Benchmark. CoRR 2016.
>
> [4] LAION-5B: An open large-scale dataset for training next generation image-text models. NeurIPS 2022.

---

> > ### Comment · Reviewer_Kpcx · 2023-08-26
> >
> > Thanks for the response. It addressed most of my concerns.

---

> > > ### Author Response · Authors · 2023-08-27
> > >
> > > Dear reviewer Kpcx,
> > >
> > > Thanks for your suggestion and reply.
> > >
> > > Best regards, Authors

---

### Official Review · Reviewer_asP3 · 2023-07-21
**Motion-X dataset together with an automatic annotation pipeline**

**Rating:** 8
**Confidence:** 5
**Correctness:** Yes, the claims are correct and the d…
**Clarity:** Yes, the paper is well written.

**Strengths:**

1. The proposed Motion-X dataset with 96K motion sequences is currently the largest text-motion datasets, which will definitely promote the research works on human motion generation;
2. The automatic pipeline for whole-body motion and text annotation is interesting and will serve as a step towards automatic text-motion generation tool.

**Additional Feedback:**

Have the authors considered to extend the single-human motion datasets to multi-persons?

**Documentation:**

Yes.

**Ethics:**

No.

**Limitations:**

Yes, the authors adequately addressed the limitations of motion quality of the annotation pipeline and the SMPL-X representations, and also the experimental setups.

**Opportunities For Improvement:**

1. The main concern is about the robustness and usability of the pipeline, since the algorithms may fail under highly occlusion, self-occlusion/viewpoints, poor illumination, low resolution etc. And no failure cases or the failure ratio are provided:

    a). Although a comprehensive pipeline includes whole-body keypoint estimation, temporal smoothing and 3D human model fitting, it is still challenging to obtain accurate and robust 3D human motions from monocular even multi-view videos;

    b). The pipeline uses Video-LLaMA to generate the video descriptions as supplemental texts, however, it cannot guarantee usable text descriptions for the hard cases, and EMOCA (side-view) and PoseScript are the same;

2. The facial expressions of GRAB dataset is filled by the BAUM dataset, which means that the Motion-X cannot reflect the real-world correspondence between human motions and facial expressions and the down-streaming generation is less meaningful;

**Relation To Prior Work:**

Yes, previous works are clearly discussed.

**Summary And Contributions:**

This paper considers the expressive whole-body human motion dataset with text annotations with two main contributions:

1. This paper proposes a large-scale 3D expressive whole-body motion dataset called Motion-X, which includes facial expressions, hand gestures and fine-grained pose descriptions and diverse scenes.
2. A whole-body motion and text annotation pipeline is also developed to automatically annotate the motion-text pairs.

The Motion-X dataset includes 96K motion sequences and 13.7M precise 3D whole-body pose descriptions. Experiment results show that the annotation pipeline and the the proposed dataset is accurate and can facilitate the research works on human motion generation and whole-body human mesh recovery.

---

> ### Author Response · Authors · 2023-08-21
> **Rebuttal by Authors**
>
> &nbsp;
> ### Response to Reviewer asP3
>
> &nbsp;
>
> Thanks for your valuable comments.
>
> &nbsp;
>
> `Q-1`: Robustness and usability of the pipeline. Example of failure cases or failure ratio.
>
> `A-1`: (a) Motion annotation: It's indeed challenging to obtain accurate 3D human motions from videos. During our experiments, we found that annotation quality correlates with video quality. Accordingly, during video collection, we rigorously filtered out low-quality videos through manual selection, human tracking, keypoints detection, and camera transition identification, as detailed in Sec. G.1 of the revised Appendix. These methods ensure most collected videos contain high-resolution, visible, and temporal consistent single person, facilitating accurate 3D motion annotation.
>
> (b) Text annotation: When using Video-LLaMA for supplementary descriptions, we input text obtained from original videos as prompts to guide the generation. These texts are derived via title extraction, OCR, or action labels from action datasets, e.g., HAA500 and HuMMan, guiding Video-LLaMA to generate usable text descriptions in most cases. For EMOCA (side-view), we use temporal correspondence by interpolating side-view facial expressions with front-view frames within the same video sequence based on visible facial keypoints numbers to make the facial estimation consistent and robust to different views. PoseScript is based on some pre-defined posecodes and may fail in some hard cases. However, as shown in Table 6, our proposed facial expressions and hand pose descriptions can enhance human motion generation, indicating that these descriptions can depict the facial expression and hand pose well and provide some useful information.
>
> (c) Failure cases: Our annotation pipeline might fail when the person resolution is too small or there exists severe truncation and occlusion. We have included visual examples in Sec. G.4 of our revised Appendix. Notably, we have performed strict data preprocessing of the collected videos and manually checked the motion labels to filter out the aforementioned cases.
>
> &nbsp;
>
> `Q-2`: The facial expressions of GRAB are filled by BAUM and cannot reflect the real-world correspondence.
>
> `A-2`: Thanks for your advice. Most motion sequences in Motion-X have facial expressions annotated from original RGB videos, reflecting real-world correspondence. GRAB constitutes a small portion (only 1.35\% in the clip number), where it provides precise body and hand annotations. We integrate expressions from the BAUM dataset to augment GRAB's facial expressions for the following reasons:
>
> (a) Our experiments revealed that most GRAB motion sequences have a neutral expression. Thus, we introduced facial expression augmentation to enhance diversity to serve and train text-to-motion generative models.
>
> (b) Besides, we find that the facial expression often decouples from the body and hand pose. For instance, 'lifting airplane happily' and 'lifting airplane neutrally' are both plausible. Thus, we perform the facial augmentation and incorporate emotion labels (e.g., happy, sad, and surprise) into the text description to ensure the consistency and correspondence between text and motion.
>
> (c) Finally, the FLAME expression in the GRAB dataset has 10 components, while the facial expression annotated by our pipeline contains 50 components to make the expression more diverse and natural. To ensure uniformity across sub-datasets, we do not use the original facial expression.
>
> However, as you suggested, this augmentation may sacrifice real-world correspondence in some activities with specific facial motions, e.g., eating an apple. Thus, we will follow your advice and use original expressions for these sequences.
>
> &nbsp;
>
> `Q-3`: Extending the single-human motion datasets to multi-persons.
>
> `A-3`: Yes, multi-person human motion capture and generation is a very promising and interesting topic. We would consider extending Motion-X for multi-person scenarios in the future. Recently, there is a two-person interaction motion dataset, InterGen [1], which provides body-only motions captured in laboratory environments.
>
> &nbsp;
>
> Reference
>
> [1] InterGen: Diffusion-based Multi-human Motion Generation under Complex Interactions. Arxiv.

---

> > ### Comment · Reviewer_asP3 · 2023-08-24
> >
> > Thanks for the clarifications. The authors have addressed most of my concerns.

---

> > > ### Author Response · Authors · 2023-08-24
> > >
> > > Dear reviewer asP3,
> > >
> > > Thanks for your reply and recognition of our work!
> > >
> > > Best regards,
> > > Authors

---

### Official Review · Reviewer_XGnA · 2023-07-21
**Review for Motion-X**

**Rating:** 7
**Confidence:** 4
**Correctness:** Yes.
**Clarity:** Yes.

**Strengths:**

1. The contribution of the Motion-X dataset is valuable to the 3D human motion generation and recovery community.

2. The automatic motion and description annotation pipeline could generate human text-motion data for online videos and is scalable. Quantitative experiments evaluate the accuracy.


**Additional Feedback:**

No.

**Documentation:**

Yes.

**Ethics:**

No. No ethics discussion is addressed.

**Limitations:**

Yes.

**Opportunities For Improvement:**

1. As the authors mentioned, the fitting accuracy is lower than the marker-based mocap systems, therefore, the annotation accuracy for head and hands may still need to be improved.

2. For indoor scenes, it seems most data are a human with only the upper body visible. Therefore, the accuracy and diversity of the lower body are to be discussed.


**Relation To Prior Work:**

Yes.

**Summary And Contributions:**

This paper proposes Motion-X, a large-scale 3D expressive whole-body motion dataset. Marker-based mocap systems constrain the collection scenarios which limits the dataset diversity and generalization ability. Therefore, Motion-X collects massive online videos and designs an automatic motion and description annotation pipeline, enabling efficient capture of high-quality human text-motion data at scale. Experiments demonstrate the effectiveness and advantages of Motion-X in whole-body motion generation and mesh recovery tasks.

---

> ### Author Response · Authors · 2023-08-21
> **Rebuttal by Authors**
>
> &nbsp;
> ### Response to Reviewer XGnA
>
> &nbsp;
>
> Thanks for your valuable comments.
>
> &nbsp;
>
> `Q-1`: Improve the annotation accuracy of head and hands.
>
> `A-1`: Thanks for your advice. It's indeed challenging to annotate precise face and hand motions, especially from low-quality videos with severe occlusion, motion blur, truncation, and low resolution.
>
> (a) We've diligently enhanced annotation accuracy by improving the input video quality and our annotation pipeline. Firstly, during our experiments, we found that annotation quality correlates with video quality.  Thus, to ensure accuracy, we rigorously filter low-quality videos by employing manual curation, human tracking, keypoint detection, and camera transition detection, as detailed in Sec. G.1 of the revised Appendix. Besides, we also trained hand and facial keypoints models on extensive datasets for greater robustness (e.g., the expert ViT-Face, ViT-Hand models from Figure 5). By fusing the advanced hand detector [1] with whole-body keypoints, we overcome hand omission issues prevalent in prior methods like OpenPose [2]. A score-guided adaptive smoothing technique further mitigated per-frame annotation jitter. Based on the above strategies, the quality of hand and face annotation can be improved greatly.
>
> (b) Notably, marker-based systems are expensive and impractical beyond labs, sometimes only capturing body motions. Hand and face estimation needs additional devices. We will keep elevating the annotation quality through innovative techniques. By consistently refining our pipeline, we aim to provide accurate annotation conveniently and cost-effectively.
>
> &nbsp;
>
> `Q-2`: Accuracy and diversity of the lower-body motion in upper-body scenes.
>
> `A-2`: Thanks for your advice. We visualize the indoor motions with only the upper-body visibility (from Figure 4 (b)) to show the robustness and effectiveness of our annotation method since most existing mesh recovery models are hard to handle in these severe scenarios. In our experiments and annotation processes, we have taken the lower-body motion accuracy into consideration and tried to improve it through data preprocess and lower-body motion augmentation.
>
> (a) During data preprocessing, we perform keypoint detection on candidate videos and discard samples with insufficient visible keypoints based on confidence scores. This method ensures most collected videos include both upper and lower body, facilitating accurate lower-body motion annotation.
>
> (b) In scenes emphasizing upper-body motion, like instrument playing, we retain individuals with upper-body visibility. For these scenes, a lower-body augmentation mechanism (Sec. B.3 in Appendix) is implemented. This involves selecting the most relevant motion from AMASS using SMPL-X parameters and substituting the lower-body motion. We also integrate relevant keywords (e.g., sitting, standing, walking) in text descriptions. AMASS guarantees accuracy and diversity in these scenes. Visual results are available in Fig. 2(a) of the Appendix. Notably, the augmentation will benefit the text-to-motion generation task because of the consistency between text description and whole-body motions. For the whole-body mesh recovery task, we simply use visibility masks to avoid the harm of invisible parts with inaccurate annotation.
>
> &nbsp;
>
> Reference:
>
> [1] Whose Hands Are These? Hand Detection and Hand-Body Association in the Wild. CVPR 2022.
>
> [2] OpenPose: Realtime Multi-Person 2D Pose Estimation using Part Affinity Fields. TPAMI 2019.

---

> > ### Comment · Reviewer_XGnA · 2023-08-28
> >
> > Thanks for the author's response. While my concerns remain, I acknowledge and appreciate the valuable contribution of the dataset. Despite the weaknesses, the strengths of the dataset outweigh them. Therefore, I maintain my original rating.

---

> > > ### Author Response · Authors · 2023-08-28
> > >
> > > Dear reviewer XGnA,
> > >
> > > Thanks for your reply and recognition of our work! We'll keep improving our annotation pipeline.
> > >
> > > Best regards,
> > >
> > > Authors

---

### Official Review · Reviewer_HxL5 · 2023-07-23
**Large-scale dataset with automatic annotation pipeline**

**Rating:** 6
**Confidence:** 4

**Strengths:**

1.Motion-X is a large-scale 3D expressive whole-body motion dataset that overcomes the limitations of existing datasets by 1) including facial expressions, hand gestures, and fine-grained pose descriptions. 2) combining different dataset and online videos for sufficient diversity and quantity 3) achieving automatic text annotation to enable large-scale dataset

2.The authors design a scalable and systematic pipeline for motion and text annotation, ensuring high precision and cost-effectiveness. The pipeline can automatically annotate motion from both multi-view and single-view videos, making it versatile for different scenarios. The experiments shows the advantages of the automatic pipeline for annotation compare with STOA methods

3.Motion-X's strength lies in its ability to enhance diverse, expressive, and natural motion generation. The dataset serves as a benchmark for evaluating state-of-the-art motion generation methods and proves valuable for the 3D whole-body human mesh recovery task, showcasing its versatility and quality for various research applications.

**Additional Feedback:**

the demo video only includes audio and not sure it is happen to all reviewers or just on my side

**Clarity:**

Yes, the paper appears well-structured and coherent, with clear explanations of the dataset creation process and the development of the annotation pipeline.

**Correctness:**

This is an efficient way to create the dataset but I am not quite sure on the quality of the dataset and potential human labor needed in this way.

**Documentation:**

Yes, the paper provides comprehensive documentation about the data collection, availability, statistic analysis and license. There is not much information about the maintenance plan.

For benchmark, it include sufficient details to reproduce the work.

**Ethics:**

I am wondering have you did some ethic analysis on the data distribution of different skin tone, gender and age? For the disabled, for example, people in a wheelchair, it might be more difficult to get accurate estimate of their motion and also the language annotation may hurt their feeling.

Have you made efforts to avoid adding harmful videos when collect videos online? For example, pornography, violence, discrimination, etc

**Limitations:**

This is a good paper which summarize all the available datasets and make them more powerful by including more annotations through automatic pipeline. Please see my question on the "opportunity for improvement" and "ethic" part.

**Opportunities For Improvement:**

I just have several questions regarding the dataset creation process:
1) How did you make sure the online video has same high quality as other datasets? Some videos may have watermark or are captured in low quality camera. Do you get the permission of those people to use their video for research purpose?
2) The automatic annotation pipeline is efficient but it cannot guarantee the quality of annotation. That's why we need human labor to go through it. How did you know which video is labeled correct or not? It seems expensive if people need to go through all the videos
3) To my understanding, it is really difficult to get the correct understanding of the hand gesture and this has great value for potential robotics research, could you show more visualization results about that annotation?

**Relation To Prior Work:**

Yes, it gives a clear discussion of how Motion-X differs from previous contributions, particularly in terms of dataset completeness, scalability, unification, and its potential to advance the field of motion generation research.

**Summary And Contributions:**

The paper presents Motion-X, a large-scale 3D whole-body motion dataset with facial expressions, hand gestures, and detailed pose descriptions. Existing datasets lack these elements and are limited by manual annotations and small-scale scenes. To address this, the authors develop a precise and scalable annotation pipeline for motion and text annotations. Using this pipeline, Motion-X is created, containing 13.7 million 3D whole-body pose annotations (SMPL-X) across 96,000 motion sequences from diverse scenes. The dataset also includes frame-level pose descriptions and sequence-level semantic labels. Experiments confirm the pipeline's accuracy and demonstrate Motion-X's value in enhancing expressive, diverse, and natural motion generation, as well as 3D whole-body human mesh recovery tasks. Overall, the contributions include the Motion-X dataset and the efficient annotation pipeline, which opens up new possibilities for research in the field of whole-body motion analysis.

---

> ### Author Response · Authors · 2023-08-21
> **Rebuttal by Authors - Part 1**
>
> &nbsp;
>
> ### Response To Reviewer HxL5 - Part 1
>
> &nbsp;
>
> Thanks for your valuable comments.
>
> &nbsp;
>
> `Q-1`: Quality of online videos.
>
> `A-1`: Ensuring online video quality is indeed a primary concern during our collection. As mentioned in Lines 101-105, we preprocess the collected online videos to filter out the low-quality videos. Here are three key strategies:
>
> (a) Manual selection for videos. When downloading the videos from online sources, we manually check the quality of the videos and eliminate the low-quality videos with multi-person scenes, low resolution, severe blur, excessive occlusion, poor illumination, etc.
>
> (b) Human tracking and keypoints detection of candidate videos. For the candidate videos, we perform human tracking and keypoints detection. To ensure a high resolution of the human region, only samples with large human bounding boxes are retained. We can automatically filter out the samples with short tracking spans or multiple individuals within a frame. Besides, we discard the person with severe truncation and occlusion based on the keypoint confidence score and only preserve the person whose visible keypoint number surpasses a predefined threshold.
>
> (c) Camera transition detection. To improve the accuracy of human tracking, shot transition detection is employed to segment videos into multiple continuous clips prior to human tracking. Clips with insufficient length are subsequently excluded.
>
> These approaches ensure the inclusion of videos with a high-resolution, continuous, single-person while effectively eliminating videos with severe truncation, occlusion, and abrupt camera transitions. As shown in the following experiments, we compare the collected online videos with existing action datasets (i.e., NTU-RGBD120, HAA500) to show its superiority from a larger person resolution, more visible keypoints, and longer clip length.
>
> |               | Person Resolution | Visible Keypoints Num | Clip Length |
> | :-----------: | :---------------: | :-------------------: | :---------: |
> | Online Videos |      285697       |          120          |     226     |
> |    HAA500     |      163715       |          79           |     61      |
> |  NTU-RGBD120  |      146298       |          108          |     68      |
>
> Table R1: Quality comparison of online videos, HAA500 and NTU-RGBD120. Online videos exhibit superior quality with a larger person resolution, more visible keypoints, and longer clip length.
>
> We have additionally conducted a user study wherein we randomly selected ten videos from each dataset and asked participants to assess and rank the quality of the three datasets. As shown in the following table, online videos are predominantly deemed to have the highest quality.
>
> |               | Highest Quality | Middle Quality | Worst Quality |
> | :-----------: | :-------------: | :------------: | :-----------: |
> | Online Videos |      66.7%      |     23.3%      |     10.0%     |
> |    HAA500     |      13.3%      |     13.3%      |     73.3%     |
> |  NTU-RGBD120  |      33.5%      |     31.8%      |     44.7%     |
>
> Table R2: User studies to compare the video quality of online videos, HAA500, and NTU-RGBD120. Online videos are predominantly deemed to have the highest quality.
>
> &nbsp;
>
> `Q-2`: Permission of online videos.
>
> `A-2`: In fact, a large portion of our videos are either authorized by original owners or collected from public platforms like Pixabay that permit commercial and academic use. However, since our dataset primarily focuses on text-driven motion generation task, we would only provide the annotated text-motion labels and would not distribute the RGB videos. If RGB videos are needed, we will follow previous works YouTube-8M [1] and LAION [2] to provide the links to the online videos for the users to download by themselves.
>
> &nbsp;
>
> Reference:
>
> [1] YouTube-8M: A Large-Scale Video Classification Benchmark. CoRR 2016.
>
> [2] LAION-5B: An open large-scale dataset for training next generation image-text models. NeurIPS 2022.

---

> > ### Author Response · Authors · 2023-08-21
> > **Rebuttal by Authors - Part 2**
> >
> > &nbsp;
> >
> > ### Response To Reviewer HxL5 - Part 2
> >
> > &nbsp;
> >
> > `Q-3`: How did we know which video is labeled correct or not?
> >
> > `A-3`: As shown in Fig. 5, our motion annotation pipeline can be divided into three steps, including keypoints estimation, local pose optimization, and global translation optimization. For each step, we identified common attributes among failed motion sequences, such as excessive keypoint jitter in keypoint estimation (evaluated by keypoint temporal deviation), or poor alignment in local pose optimization (evaluated by 2D/3D keypoints fitting error). Thus, after annotation, we calculate the temporal deviation and keypoint fitting error of the motion sequences for preliminary screening, and verify the validity of the motion sequences with notable jitter or fitting error through the following steps:
> >
> > (a) Project the body mesh onto the image plane, overlaying it on the original RGB image for 2D alignment inspection.
> >
> > (b) Utilize Blender to visualize the human motion in world coordinates, examining global trajectory plausibility and continuity.
> >
> > Motion sequences exhibiting coherent global trajectories and image alignment are labeled as valid, while others are deemed invalid.
> >
> > Notably, we validate through visualization that the automatic filtering using temporal deviation or projection error exhibits a high level of consistency with human assessment, highlighting the effectiveness and efficiency of this automated preliminary screening process.
> >
> > &nbsp;
> >
> > `Q-4`: More visualization results of hand gesture text annotation.
> >
> > `A-4`: Describing hand gestures is challenging since the same gesture may have different descriptions depending on the application context. To comprehensively describe the hand gesture, we provide both low-level pose descriptions for individual frames and high-level semantic labels for entire motion sequences:
> >
> > (a) For semantic labels, we refer to existing hand-object interaction (HOI) and hand gesture recognition datasets and employ some atomic verbs to describe the hand gesture (e.g., fist clenching, handshake). These atomic verbs are relatively limited in number and easy to annotate.
> >
> > (b) For pose description, we extend the posecodes of PoseScript to encompass hand gestures, resulting in HandScipt, which primarily captures the pose state of each finger (e.g., the left thumb is bent a bit) and the physical relationship among fingers (e.g., the right thumb is spread apart from other fingers) based on the hand pose and keypoints position.  We have added some visualization results of the HandScript in Sec. G.3 of our revised Appendix.
> >
> > It's very interesting and valuable to combine our method with robotic research. We will incorporate additional robotic-related atomic verbs and pose codes in the future.
> >
> > &nbsp;
> >
> > `Q-5`: Ethics concern.
> >
> > `A-5`: Thanks for your kind reminder.
> >
> > (a) As mentioned in `A2`, we will provide the text-motion annotations and would not distribute the RGB videos. Meanwhile, in our provided texts, we do not describe the appearance but only motion information. Thus, it would not involve shape-related factors such as skin tone, gender, and age.
> >
> > (b) For the disabled, we will use the large language model to select relevant text descriptions and then meticulously validate through manual review.
> >
> > (c) During video collection, we have restricted the scenarios to exclude harmful videos. Besides, we use some pretrained filters such as Jigsaw to detect the harmful content in the text labels and employ NSFW from LAION [2] to filter out the pornography videos.
> >
> > &nbsp;
> >
> > `Q-6`: Demo video only includes audio.
> >
> > `A-6`: We have double-checked the demo video and there exists both RGB video and audio. Please try to change a video player or refer to our project website: https://motion-x-dataset.github.io.
> >
> > &nbsp;
> >
> > Reference:
> >
> > [1] YouTube-8M: A Large-Scale Video Classification Benchmark. CoRR 2016.
> >
> > [2] LAION-5B: An open large-scale dataset for training next generation image-text models. NeurIPS 2022.

---

> > > ### Comment · Reviewer_HxL5 · 2023-08-23
> > >
> > > Thanks for the clarifications. The author addressed most of my concerns.

---

> > > > ### Author Response · Authors · 2023-08-24
> > > >
> > > > Dear reviewer HxL5,
> > > >
> > > > Thanks for your suggestions and reply.
> > > >
> > > > Best regards,
> > > > Authors

---

### Decision · Program_Chairs · 2023-09-22

**Decision:**

Accept (Poster)

**Comment:**

This submission introduces a large-scale 3D whole-body motion dataset, which consists of GT from motion capture datasets (e.g., AMASS) and pseudo-GT from in-the-wild videos. One of the major bottlenecks for 3D whole-body human motion estimation is a lack of dataset, which makes this work highly useful for the community. There were some unclear and misleading descriptions in the original submission, for example, do not clearly discriminate GT from motion capture datasets and pseudo-GT from in-the-wild videos. However, in the rebuttal, the authors clearly addressed this concern and promised to revise the manuscript to clearly discriminate against them. Given positive reviews from all reviewers, AC recommends acceptance of this paper.